# Protective and risk factors of mental health of working age adults with adventitious total bilateral blindness and low vision: A scoping review protocol

Nneoma Dike[1,2]*, Lucia D'Ambruoso[3,4,5,6,7], Heather May Morgan[2], Zoë Skea[8], Emma-Louise Tarburn[3]

1 Department of Ophthalmology, Rivers State University Teaching Hospital, Port Harcourt, Rivers State, Nigeria, 2 Institute of Applied Health Sciences, University of Aberdeen, Aberdeen, Scotland, United Kingdom, 3 Aberdeen Centre for Health Data Science, Institute of Applied Health Sciences, School of Medicine, Medical Sciences and Nutrition and Centre for Global Development School of Education, University of Aberdeen, Aberdeen, Scotland, United Kingdom, 4 Department of Epidemiology and Global Health, Umeå University, Umeå, Sweden, 5 School of Public Health, University of the Witwatersrand, Johannesburg, South Africa, 6 Public Health Directorate, National Health Service, Grampian, United Kingdom, 7 Department of Global Health, Centre of Global Surgery, Stellenbosch University, Stellenbosch, South Africa, 8 Health Services Research Unit, University of Aberdeen, Aberdeen, Scotland, United Kingdom

☯ These authors contributed equally to this work.
* n.dike.22@abdn.ac.uk

**Data Availability Statement:** No datasets were generated or analysed during the current study. All

## Abstract

Vision loss has been associated with mental health problems such as depression, anxiety, and post-traumatic stress disorder, which significantly impact lives of working age adults with adventitious total bilateral blindness and low vision. It is imperative, therefore, to prioritize the mental health in this population by exploring and understanding the factors that impact on their mental health. Hence, the objective of this scoping review is to identify and chart existing literature on the protective and risk factors of mental health of working age adults with adventitious total bilateral blindness and low vision. We developed this scoping review protocol in line with the Joanna Briggs Institute guidance. This scoping review will include publications in English language with no date restrictions exploring the protective and risk factors of mental health of our study population. A three-step search strategy will be employed. Searches will be carried out in the following databases: Medline, Embase, PsycInfo, PsycArticles, CINAHL and Web of Science. Search for grey literature will be conducted in Google, Google Scholar and Websites dedicated to information on visual impairment. Collated results will be imported into Endnote Basic (Clarivate) for deduplication. Two reviewers will independently conduct double screening of all the titles and abstracts in Rayyan- a web application, and full texts in Endnote while three other reviewers will conduct screening of a subset of for example 10% of titles and abstracts and full texts. Furthermore, two reviewers will independently conduct double data extraction while three other reviewers will revise, cross check, and correct any extraction errors. Extracted data will be presented in tabular formats and summarized descriptively in line with the research objectives. This scoping review will generate evidence on factors impacting the mental

relevant data from this study will be made available upon study completion.

**Funding:** The author(s) received no specific funding for this work.

**Competing interests:** The authors have declared that no competing interests exist.

health of the working age adults with adventitious total bilateral blindness and low vision as well as critically highlight gaps in the literature. The findings will inform and critically underpin future empirical research which will explore the lived experiences of working age people with adventitious total bilateral blindness. Additionally, evidence from this review will inform the development of interventions in the promotion of mental health as well as assisting rehabilitation specialists and workers, public health practitioners and other relevant stakeholders in addressing the mental health needs of working age adults with adventitious total bilateral blindness and low vision.

## Introduction

A human's most highly developed sense is sight [1]. Therefore, people probably rely more on vision than all other senses combined [2]. Vision is a critical tool used to engage in everyday activities, attain self-gratification, perceive beauty, mobility, learn, utilize tools, and enjoy every aspect of life [1]. Predictably, a cross-sectional study conducted in the United Kingdom showed that the most valued human sense is sight [3]. Consequently, vision loss has a significant negative impact on everyday activities, depressive symptoms, and feelings of anxiety [4]. Moreso, individuals with vision loss commonly have depression [5] and post-traumatic stress disorder [6]. Additionally, a cross-sectional study showed that people with loss of vision had 4.6 times significantly greater risk of experiencing psychological distress than people with normal sight [7]. Critically, findings from a more recent comparative cross-sectional study, showed that psychological distress was significantly more prevalent in visually impaired adults than adults with normal vision [8]. More importantly, depression is linked to poorer outcomes in vision rehabilitation [9]. Even foreseeable vision loss can cause severe psychological suffering that could result in suicide [10]. Conclusions from a systematic review showed that vision loss usually impacts negatively on the quality of life as well as the mental health [11].

Vision loss, also called visual impairment, can be congenital (present at birth or before the age of 5) or adventitious (acquired at or above the age of 5) [12, 13]. According to Bruce [12], the impact of visual impairment on people, depends on the age at onset. Findings from a cross-sectional study showed significantly greater levels of depression in students with adventitious visual impairment compared to those with congenital visual impairment [14]. Moreso, the people with adventitious visual impairment have poorer quality of life and find it more difficult to deal with vision loss than those with congenital visual impairment [15]. Considering the evidence that the mental health of people with adventitious visual impairment is disproportionately impacted compared to that of those with congenital visual impairment, the focus of this scoping review is adventitious visual impairment which encompasses both total bilateral blindness and low vision [15].

Considering the mental health problems encountered by people with adventitious vision loss, prioritizing their mental health becomes critical. However, to optimally achieve prioritization, it is imperative that factors impacting on their mental health are comprehensively explored and understood. Hence, this scoping review will explore the protective and risk factors of mental health of people with adventitious visual impairment. Protective and risk factors are defined as the diverse and usually complex factors impacting on mental health and well-being [16]. On one hand, protective factors are those factors which strengthen the mental health of an individual and operate in ways that will improve the ability of an individual to cope with problematic situations [16]. On the other hand, risk factors have adverse impact on

the mental health of an individual [16]. Essentially, protective and risk factors aid in explaining the reason for the form and nature, distribution, and persistence of a problem [17]. To achieve and sustain positive mental health, protective factors need to be developed, risk factors minimized and barriers to seeking help, removed [16].

It is, however, imperative for the authors to elucidate the meaning of protective and risk factors for mental health within the context of this scoping review. Critical findings from studies have established that certain factors positively or negatively impact on mental health. These factors will be considered as protective or risk factors of mental health respectively in this scoping review. For instance, multidisciplinary rehabilitation (including visual rehabilitation) [18–20] and physical activity [21] improve mental health of adults and therefore will be considered as protective factors.

Conversely, vision-specific distress which refers to the type of distress associated with coping with visual impairment has been found to be a critical risk factor for depressive symptoms [22]. In two cross-sectional studies involving visually impaired adults, vision-specific distress compared to other variables like social support and coping, was found to have the most unique and strongest association with depression [22, 23]. Although higher levels of general coping tendencies as adaptive mechanisms have been associated with lowered depressive symptoms in people with visual impairment [24], avoidance coping however, is significantly associated with depressive symptoms making it a risk factor for mental health problems [22, 25].

Additionally, a life of chronic disability frequently consists of dependence on friends and family for assistance with emotional support and instrumental tasks [26]. Such social support has been acknowledged by researchers as having potential to positively affect the lives of persons with chronic disability [26]. It has however been acknowledged in recent times, that there are negative aspects of social support like being angry and hostile which are capable of adversely impacting the lives of people with chronic disability [26, 27]. Hence, social support can comprise of positive and negative aspects that exist concurrently in a person's environment [26]. According to Antonucci, Lansford and Akiyama [27], close friends and significant others can be sources of affection and love as well as antagonism and distress. In their research, positive and negative facets of social support impacted on well-being of the study participants differently. For instance, depressive symptomatology was significantly correlated with significant others, getting on one's nerves (negative social support) while life satisfaction was significantly associated with confiding in spouse (positive social support) [27]. Moreover, the findings from a systematic review showed that spousal support as a source of social support was the most consistent significant protective factor against depression for adults [28]. In this present scoping review therefore, positive social support will be considered a protective factor of mental health while negative social support will be taken as a risk factor for mental health problems.

Furthermore, research has identified stigmatization as detrimental to mental health. According to Thornicroft et al [29], stigma is a symbol of discredit which evokes negative attitudes directed towards the stigmatized. It is an all-encompassing term which consists of discriminatory acts, lack of knowledge and prejudice [29]. They argue that negative thoughts are not the only acts of prejudice that the majority exhibit in the rejection of the minority, but they can also be hostile, resentful, anxious, angry, and repulsed [29]. Critically, Link and Phelan [30] assert that individuals are considered stigmatized when they lose status and are discriminated against because of being labelled, separated, and associated with repugnant attributes. Generally, stigmatized groups are at a disadvantage in terms of life chances such as mental health, earnings, education, health, accommodation, status, and medical care [30]. According to Williamson [31], social stigma can aggravate depression as well as development of self-stigmatization which is conceptualized as having negative belief about oneself (stereotype),

agreeing with said belief (prejudice) and responding to prejudice (discrimination) [32]. In a study conducted by Van Munster *et al* [33], participants believed that their mental health issues were exacerbated when they felt vulnerable and unequal because of their visual impairment. Many people with visual impairment face self-stigma with regards to the visual impairment as well as mental health issues which can aggravate these problems [33]. Therefore, stigmatization is considered a risk factor for mental health problems.

According to the World Health Organization (WHO), people with disabilities are twice as likely to develop depression [34]. The WHO asserts that such inequalities stem from social situations encountered by people with disabilities such as stigmatization, being excluded from employment and education, discrimination, and poverty [34]. Predictably, individuals with visual impairment, a lot of whom have low socioeconomic status, face an array of challenges in being able to access sufficient healthcare which can worsen the state of their mental health [35]. Therefore, socioeconomic status (SES) is consistently and reliably predictive of mental health [36].

These examples are by no means exhaustive. A wide range of factors exist and will be explored by this scoping review.

Importantly, exploring protective and risk factors for mental health problems among working age adults with adventitious total bilateral blindness and low vision will be critical in the promotion of mental health literacy for service users and providers. According to O'Connor, Casey and Clough [37] and Jorm *et al* [38], knowledge of factors associated with mental health problems is a major attribute of mental health literacy. Moreso, while the significance of health literacy for physical health is broadly recognized, the field of mental health literacy is comparatively neglected [39, 40]. Given that this review will explore and chart factors impacting on mental health problems, it will boost knowledge and create awareness of these determinants thereby promoting mental health literacy for service users and providers.

Scoping reviews are appropriate for mapping relevant evidence on a specific subject matter or field of research and support for the identification of fundamental concepts, research gaps and sources and kinds of evidence for informing research, practice and making of policies [41]. Given that scoping reviews are conducted for varied reasons, it is imperative to clearly explain the purpose for which this scoping review is being conducted [42]. According to Peters *et al* [43], scoping reviews are used to assess and understand the scope of the evidence in an emerging area or for the identification, charting, reporting or discussion of the features/concepts in that area. Hence, they are inherently exploratory [44]. A scoping review was therefore, considered to be the most appropriate type of review to meet the objectives of the proposed research. This is because, it will allow the researchers to explore, identify, chart, and report available evidence on potential factors to promote positive mental health (protective factors) and factors that might be detrimental to positive mental health (risk factors) of working age adults with adventitious total bilateral blindness and low vision.

Importantly, Møller [45 p. 32] asserts that a way of "mapping" the evidence to unravel what is known and not known is to conduct systematic reviews. It has been established that mapping, as a term, is frequently utilized when explaining a process of evidence analysis [46]. However, Khalil *et al* [46] argue that it may be the case that the meaning of mapping and how it is attained may be misconstrued. They clarify that mapping can be used to ascertain the scope of a scale of evidence in a specific area [46]. The "mapping" in our scoping review aligns with Khalil *et al*'s [46] clarification which is to scope the extent of available evidence on protective and/or risk factors of mental health for our target population.

The conduct of our scoping review will be underpinned by methodological rigor and transparency throughout. In addition, dissemination of research findings and informing future research (for instance systematic reviews, primary research, and other forms) are among the

core expectations and outputs of properly conducted scoping reviews [47]. Hence, this scoping review will be pivotal in making contributions to empirical research and appropriate recommendations which will be underpinned by the review's findings.

A preliminary search of Prospero (22nd March 2023), Cochrane Database of Systematic Reviews (22nd March 2023) and JBI Evidence Synthesis (26th March 2023) did not identify any published or ongoing review (scoping or systematic) on this topic. Some reviews like Senra *et al* [11] have studied the impact of adventitious vision loss on mental health, however, the focus of our review will go beyond that to explore the determinants that can maintain the integrity of the mental health as well as those that can deteriorate mental health in our study population. Moreover, Senra *et al*'s [11] review was conducted almost a decade ago with no major advances in the field, further justifying the critical need for this scoping review. Hence, conducting this scoping review is timely and of critical importance as it will fill the gap in literature.

## Methods

This proposed scoping review will be carried out in accordance with the latest Joanna Briggs Institute (JBI) methodology for scoping reviews [42, 48] and the PRISMA-ScR reporting guidelines (see S3 Appendix), both of which underscore the significance of methodological rigor [49]. Also, evidence and knowledge gaps will be identified, and appropriate recommendations for service provision made.

### Research question

What is known from existing literature about the protective and risk factors of mental health of working age adults with adventitious total bilateral blindness and low vision?

### Research sub questions

1. What are the protective factors of mental health for people with adventitious total bilateral blindness and low vision?

2. What are the risk factors for mental health problems for people with adventitious total bilateral blindness and low vision?

3. What are the evidence gaps related to mental health among working age adults with adventitious total bilateral blindness and low vision?

### Inclusion criteria

**Participants.** The study population is working age adults aged 18–65 years who have adventitious total bilateral blindness or low vision. Studies that include participants with congenital blindness, other types of sensory impairment, multiple morbidities or aged above or below the age group 18–65 years will not be considered.

**Concept.** This scoping review will consider quantitative studies, qualitative studies and mixed methods studies and grey literature published in English language that explored the protective and/or risk factors of mental health of working age adults with adventitious total bilateral blindness or low vision.

**Context.** Additionally, mental health and well-being of people have been known to be significantly impacted by socioeconomic status, geographic location, gender as well as race and these are well documented in the evidence base [50]. This scoping review will consider eligible

quantitative studies, qualitative studies and mixed methods studies conducted in any country and for every gender and race. This is to ensure comprehensiveness and a wider reach in this review.

## Types of sources

All types of quantitative, qualitative, and mixed methods study designs published in English language are eligible for inclusion. However, considering that no reviews on this topic were found, systematic and scoping reviews will not be eligible for inclusion. Also, commentaries, editorials, letters, conference abstracts, text and opinion papers will be excluded.

## Information sources

We will search the following databases: Medline, Embase, CINAHL, Web of Science, PsycInfo and PsycArticles. Also, to ensure comprehensiveness in sourcing for relevant articles, search for grey literature will be conducted across Google, Google Scholar as well as websites known for their work with people with visual impairment such as World Vision International, World Blind Union, Royal National Institute of Blind People (RNIB), the American Foundation for the Blind (AFB), Anglo-Nigerian Welfare Association for the Blind and websites of other relevant institutions working on visual impairment.

## Search strategy

In line with JBI guidance, a three-step search strategy will be used [42]. Step one: a preliminary limited search was conducted on the 28[th] of March 2023 in Medline, CINAHL, Embase and PsycArticles for the identification of articles on the subject matter. This first step was conducted by the review team with the help of a librarian. In this step, text words found in the title and abstract of articles retrieved as well as index terms were analysed repeatedly and subsequently refined to ensure that the search strategy has captured relevant articles. The full search strategy for the Medline database can be seen and depicted as S1 Appendix in this protocol. Step two: second search will be conducted in which all the keywords and index terms found will be used across the remaining databases: Embase, CINAHL, Web of Science, PsycInfo and PsycArticles. Step three: additional relevant studies will be sourced from searching the reference lists of all the included studies for the review.

Additionally, hand searching [51] will be conducted across relevant journals like the Journal of Visual Impairment and Blindness which is a publication by AFB Press. Also, author searching will take place whereby the name of an author(s) of a relevant article will be searched to retrieve any additional relevant article(s) that he/she/they may have written [51]. Some databases may require the search strategy to be adapted to get optimal results. Aiming for robust and comprehensive results, neither date limit nor geographic location limit will be imposed in this review. Also, we will set up key word and Table of Contents (TOC) email alerts in Google Scholar and key journals such as the Journal of Visual Impairment and Blindness and Visual Impairment Research. This proposed review will not have any date restriction. However, eligible studies will be limited to only those published in English language as this is the international language the reviewers are familiar with.

## Study/source of evidence selection

Collated search results from all the databases will be uploaded to Endnote Basic (Clarivate) where duplicates will be removed. After deduplication, title and abstract screening will be conducted independently by the review team using Rayyan- a web tool used to screen articles [52].

Two reviewers will independently double screen the titles and abstracts of the search results in line with the inclusion criteria of the scoping review while three other members of the review team will screen a subset of for example, 10% of the titles and abstracts. Articles that qualify for full text screening will then be exported back into Endnote for screening. Then, two reviewers will independently conduct double screening of full text of all the articles while three other reviewers will conduct full text screening of a subset of for example 10%. Resolution of any difference in opinion during the screening process will be through discussions or the intervention of another reviewer. The search results as well as the inclusion and exclusion process will be documented in the scoping review and presented in the PRISMA-ScR flow diagram [49].

## Data extraction

The data extraction template from the JBI Evidence Synthesis manual [42] has been adapted for this scoping review (see S2 Appendix). According to Pollock *et al* [53], only data items that have relevance to the research questions should be extracted by authors of scoping reviews. Hence, the headings of the data extraction table include title, first author and year of publication, country of publication, aims, methods, age, sex, number of participants, type of vision loss (total blindness and/or low vision), key findings (protective and/or risk factors).

Combined inductive and deductive approaches will be used during data extraction. The inductive approach will accommodate and map potential unanticipated themes while the deductive approach will entail using predetermined data items in the data extraction template to retrieve information relevant in answering our research questions. Pollock *et al* [53] argue that data extraction is an iterative process; hence, authors of scoping review may identify additional relevant data items while extracting data. New themes may arise during data extraction which may also be relevant in answering our research questions but were not part of the predetermined data extraction framework. Any additional data item extracted that was not prespecified in this protocol will be documented in the final report [53]. Two reviewers will independently conduct double data extraction which will be revised, cross checked and any data extraction errors corrected by three other members of the research team. Additionally, authors of included papers will be contacted to request for additional or omitted information should the need arise [42]. At this protocol phase, the pilot step was undertaken by two reviewers independently, who trialled the data extraction form on four different articles chosen at random to refine the template [42].

## Assessment of methodological quality

Scoping reviews unlike conventional systematic reviews do not normally assess the methodological quality or risk of bias of included studies [48]. This is because the aim of scoping reviews is not to produce results that have undergone critical appraisal or synthesis, rather, they are targeted at giving an overview of the evidence [54]. This scoping review's focus is not in answering questions of effectiveness for which the assessment of methodological quality would have been mandatory [55]. Rather, our scoping review, in line with our objectives, is exploratory and designed to give an overview of available literature on the factors impacting on mental health irrespective of quality [55]. Moreso, Pollock *et al* [53] argue that as a matter of principle, only data items which are of relevance to the objectives should be extracted in the review. Given that our methodology is such that the quality of the included studies will not play any significant role in the achievement of objectives (identification of factors impacting on mental health and knowledge gaps), assessment of methodological quality or risk of bias of included studies will not amount to best practice and therefore will not be performed.

## Data analysis and presentation

Extracted data will be presented in tabular format. However, if additional diagrammatic representation is deemed significant, then this will be considered. Thereafter, a detailed descriptive summary of the extracted data will be performed [42]. These will be interpreted in relation to the scoping review objectives. Evidence and knowledge gaps will be identified and used to develop recommendations for future research.

## Discussion

The intended scoping review aims to map the available evidence on the protective and risk factors of mental health problems of working age adults with adventitious total bilateral blindness and low vision. Additionally, the proposed review intends to report gaps in the evidence base in relation to the mental health of this target population. Moreso, to the best of the researchers' knowledge, no review to date has explored protective and risk factors of mental health of this target population. Hence, the review will add and enrich the evidence base on the determinants of mental health of this target population. The findings from this scoping review will inform the design of interventions for the study population and for the promotion of mental health literacy. Also, this review will: inform and underpin future research on the mental health of this study population; and be of assistance to rehabilitation specialists and workers, health promotion practitioners and other relevant stakeholders in tackling the mental health needs of working age adults with adventitious total bilateral blindness and low vision.

This scoping review's limitation is that it will exclude studies that are not written in English language.

## Supporting information

**S1 Appendix. Full search strategy for Medline database.**
(DOCX)

**S2 Appendix. Data extraction template.**
(DOCX)

**S3 Appendix. Preferred Reporting Items for Systematic reviews and Meta-Analyses extension for Scoping Reviews (PRISMA-ScR) checklist.**
(DOCX)

## Acknowledgments

**Ochea Ikpa:** Ensured that the corresponding author dotted every "I" and crossed every "T".

**Melanie Bickerton:** The authors collaborated with her in the development and refinement of the search strategy at the protocol stage.

**Dr. Oghenebrume Wariri:** The corresponding author received invaluable professional guidance from him throughout the development of the review protocol.

**Clare Robertson:** The corresponding author received expert guidance from her during the development of the review protocol.

**Professor Amudha Poobalan:** The corresponding author was expertly guided by her in the development of the search strategy for this review protocol.

## Author Contributions

**Conceptualization:** Nneoma Dike, Lucia D'Ambruoso, Heather May Morgan, Zoë Skea.

**Data curation:** Nneoma Dike, Lucia D'Ambruoso, Heather May Morgan, Zoë Skea, Emma-Louise Tarburn.

**Methodology:** Nneoma Dike, Lucia D'Ambruoso, Heather May Morgan, Zoë Skea.

**Project administration:** Nneoma Dike.

**Resources:** Nneoma Dike, Lucia D'Ambruoso, Heather May Morgan, Zoë Skea, Emma-Louise Tarburn.

**Supervision:** Lucia D'Ambruoso, Heather May Morgan, Zoë Skea.

**Writing – original draft:** Nneoma Dike.

**Writing – review & editing:** Nneoma Dike, Lucia D'Ambruoso, Heather May Morgan, Zoë Skea, Emma-Louise Tarburn.

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
