## [Decision Letter · Decision Letter 0]

25 Aug 2023

PONE-D-23-19617Protective and risk factors of mental health of working age adults with adventitious total bilateral blindness and low vision: a scoping review protocolPLOS ONE

Dear Dr. Dike,

Thank you for submitting your manuscript to PLOS ONE. I sincerely apologisefor the unusually delayed review timeframe. After careful consideration, we feel that it has merit but does not fully meet PLOS ONE’s publication criteria as it currently stands. Therefore, we invite you to submit a revised version of the manuscript that addresses the points raised during the review process.

We look forward to receiving your revised manuscript.

Kind regards,

Emily Chenette

Editor in Chief

PLOS ONE

Reviewers' comments:

Reviewer's Responses to Questions

**Comments to the Author**

1. Does the manuscript provide a valid rationale for the proposed study, with clearly identified and justified research questions?

Reviewer #1: Yes

2. Is the protocol technically sound and planned in a manner that will lead to a meaningful outcome and allow testing the stated hypotheses?

Reviewer #1: Partly

3. Is the methodology feasible and described in sufficient detail to allow the work to be replicable?

Reviewer #1: Yes

4. Have the authors described where all data underlying the findings will be made available when the study is complete?

Reviewer #1: No

5. Is the manuscript presented in an intelligible fashion and written in standard English?

Reviewer #1: Yes

6. Review Comments to the Author

You may also provide optional suggestions and comments to authors that they might find helpful in planning their study.

Reviewer #1: This scoping review protocol addresses and important topic and has the potential to build on the evidence-base and our understanding of the of the protective and risk factors of mental health, among working age adults with adventitious total bilateral blindness and low vision.

I have several concerns:

1) Failing to carry out some assessment of methodological quality of the included studies is a fundamental flaw of your protocol. While I understand this is not a requirement for Scoping Reviews, how can the authors draw meaningful conclusions and identify recommendations for future research if the findings are based on poorly designed studies? For example, in your discussion you state ‘your study aims to map the available evidence’. I would encourage some assessment of the quality of the included studies.

2) How will protective and risk factors be defined and identified? These are broad terms that can relate to social relationships, stigma, relationship with health care professionals etc. Some explanation of what will be considered ‘protective’ and ‘risk’ factors in the protocol would be useful.

3) Will vision-specific distress be included in the ‘mental health’ search? (e.g. see Rees et al. Investigative ophthalmology & visual science. 2013 54(12):7431-8).

4) The authors reference the Senra et al.2015 systematic review. However, acknowledgement that there have been no real advances in this area for almost a decade, should be mentioned in the introduction - as this provides further justification for the need for this scoping review.

Minor comments:

Introduction, line 5: ‘More so, individuals with vision loss have more prevalence of depression [5] and post-traumatic stress disorder [6]’ - compared with who?

Introduction, line 63: Additionally, a cross-sectional study showed that people with loss of vision had 4.6 times significantly greater risk of experiencing psychological distress than people with normal sight[7].’ - Is there any recent literature to support this?

Introduction, line 65: ‘More importantly, depression is linked to poorer outcomes in rehabilitation [8].’ - in ‘vision’ rehabilitation?

Introduction, line 80: ‘Furthermore, loss of vision is often accompanied by decreased access to information, mobility, self-regulating abilities, confidence to go out by oneself which results in social isolation, poor physiological and mental health…..’ – this paragraph seems out of place. The consequences of vision loss are discussed in your first paragraph. What is the point you are making. Is this needed?

Introduction, line 109: ‘Scoping review was therefore, considered to be the most appropriate type of review to meet the objectives of the proposed research’ – ‘A’ scoping review? Word missing.

Introduction, line 111: ‘This is because, it will allow the researchers to explore, identify, chart, and report the available evidence on the factors that might help to ‘promote positive’ mental health (protective factors). Good mental health isn’t just merely the absence of mental health problems. We need to move away from this idea that we simply need to reduce poor mental health, but rather also promote protective factors which enhance positive mental health.

7. PLOS authors have the option to publish the peer review history of their article (what does this mean?). If published, this will include your full peer review and any attached files.

Reviewer #1: **Yes: **Dr Edith Holloway

---

## [Author Response · Author response to Decision Letter 0]

2 Oct 2023

Peer Reviewer’s Comments 

1. Failing to carry out some assessment of methodological quality of the included studies is a fundamental flaw of your protocol. While I understand this is not a requirement for Scoping Reviews, how can the authors draw meaningful conclusions and identify recommendations for future research if the findings are based on poorly designed studies? For example, in your discussion you state ‘your study aims to map the available evidence’. I would encourage some assessment of the quality of the included studies.

Authors’ Comments 

Thank you for drawing our attention to this critical point. Our scoping review’s focus is not in answering questions of effectiveness for which the assessment of methodological quality would have been mandatory. Rather, our scoping review, in line with our objectives, is exploratory and designed to give an overview of available literature on the factors impacting on mental health irrespective of quality. 

The methodology of our scoping review is such that the quality of the included studies will not play any significant role in the achievement of our study’s objectives (identification of factors impacting on mental health and knowledge gaps). Considering that, only data items which are of relevance to the scoping review’s objectives should be extracted by authors of the scoping review, proceeding to assess methodological quality of included studies may not be best practice. 

We have responded to this observation in the manuscript on pages 6-7; 11 on lines 169-171 and 180-187; 310-318 respectively. 

Peer Reviewer's Comments

2. How will protective and risk factors be defined and identified? These are broad terms that can relate to social relationships, stigma, relationship with health care professionals etc. Some explanation of what will be considered ‘protective’ and ‘risk’ factors in the protocol would be useful. 

Authors’ Comments

Thank you for this important observation. We have explained in detail and given examples of protective and risk factors of mental health of our target population in the context of our scoping review. 

These corrections can be seen on pages 4-6 and lines 95-155.

Peer Reviewer's Comments

3. Will vision-specific distress be included in the ‘mental health’ search? (e.g. see Rees et al. Investigative ophthalmology & visual science. 2013 54(12):7431-8). 

Authors’ Comments

Thank you for the useful comment. This term has been incorporated into the synonyms for risk factors.

We discussed vision-specific distress on page 4 and lines 102-106.

Peer Reviewer's Comments

4. The authors reference the Senra et al.2015 systematic review. However, acknowledgement that there have been no real advances in this area for almost a decade, should be mentioned in the introduction - as this provides further justification for the need for this scoping review. 

Authors’ Comments

Thank you for your observation. We agree that acknowledging that no recent advancements in this area has been made, is also a strong justification for this research. This information strengthens our argument that this scoping review is of timely importance. 

We have made this justification on page 7 and lines 201-204.

Peer Reviewer's Comments

5. Introduction, line 5: ‘More so, individuals with vision loss have more prevalence of depression [5] and post-traumatic stress disorder [6]’ - compared with who? 

Authors’ Comments

Thank you for the useful comment. The authors’ intention was not to make comparison at this point rather we meant that depression and anxiety are common/prevalent in individuals with vision loss. We have paraphrased the sentence to reflect our intent in this part of the manuscript. 

The corrected and revised statement can be found on page 3 and lines 61-62.

Peer Reviewer's Comments

6. Introduction, line 63: Additionally, a cross-sectional study showed that people with loss of vision had 4.6 times significantly greater risk of experiencing psychological distress than people with normal sight[7].’ - Is there any recent literature to support this? 

Authors’ Comments

Thank you for raising this critical point. We have cited more recent literature which found that visually impaired people experience higher psychological distress than normal sighted adults. 

We have made this addition to our protocol on page 3 and lines 64-66.

Peer Reviewer's Comments

7. Introduction, line 65: ‘More importantly, depression is linked to poorer outcomes in rehabilitation [8].’ - in ‘vision’ rehabilitation? 

Authors’ Comments

Thank you for suggesting that we add this word as it gives the sentence a more focused and meaningful outlook. 

We have added the word “vision” on page 3 and line 67.

Peer Reviewer's Comments

8. Introduction, line 80: ‘Furthermore, loss of vision is often accompanied by decreased access to information, mobility, self-regulating abilities, confidence to go out by oneself which results in social isolation, poor physiological and mental health…..’ – this paragraph seems out of place. The consequences of vision loss are discussed in your first paragraph. What is the point you are making. Is this needed? 

Authors’ Comments

Thank you for drawing our attention to this paragraph and asking pertinent questions about its usefulness. We agree that this paragraph is out of place and therefore not needed as it adds ambiguity rather than clarity to the protocol. 

We have removed this paragraph from the manuscript. 

Peer Reviewer's Comments

9. Introduction, line 109: ‘Scoping review was therefore, considered to be the most appropriate type of review to meet the objectives of the proposed research’ – ‘A’ scoping review? Word missing. 

Authors’ Comments

Thank you for this detailed proofreading. Although it is one word, it makes a positive difference to the sentence and has better appeal with the word added. 

We have added the word “A” to the manuscript on page 6 and line 174.

Peer Reviewer's Comments

10. Introduction, line 111: ‘This is because, it will allow the researchers to explore, identify, chart, and report the available evidence on the factors that might help to ‘promote positive’ mental health (protective factors). Good mental health isn’t just merely the absence of mental health problems. We need to move away from this idea that we simply need to reduce poor mental health, but rather also promote protective factors which enhance positive mental health. 

Authors’ Comments

Thank you for raising this critical awareness in this manuscript. We agree that researchers, service providers and users need to make the transition from fixating on reduction of poor mental health to actively promoting protective factors for the enhancement of positive mental health. 

We have deleted “poor” and replaced it with “promote positive” as well as “be detrimental to positive”.

This can be found on page 6 and lines 176-177.

Peer Reviewer's Comments

11. Have the authors described where all data underlying the findings will be made available when the study is complete?

The PLOS Data policy requires authors to make all data underlying the findings described in their manuscript fully available without restriction, with rare exception, at the time of publication. The data should be provided as part of the manuscript or its supporting information or deposited to a public repository. For example, in addition to summary statistics, the data points behind means, medians and variance measures should be available. If there are restrictions on publicly sharing data—e.g. participant privacy or use of data from a third party—those must be specified.

Reviewer #1: No 

Authors’ Comments

We apologise for this omission. We have taken steps to clearly state that all data underpinning our scoping review findings will be available in the manuscript or in the supporting file when the study is completed.

We have made the above declaration of data availability on page 13 and lines 360-362.

---

## [Decision Letter · Decision Letter 1]

18 Dec 2023

Protective and risk factors of mental health of working age adults with adventitious total bilateral blindness and low vision: a scoping review protocol

PONE-D-23-19617R1

Dear Dr. Dike,

We’re pleased to inform you that your manuscript has been judged scientifically suitable for publication and will be formally accepted for publication once it meets all outstanding technical requirements.

Kind regards,

Muhammad Shahzad Aslam, Ph.D.,M.Phil., Pharm-D

Academic Editor

PLOS ONE

Additional Editor Comments (optional):

Reviewers' comments:

Reviewer's Responses to Questions

**Comments to the Author**

1. Does the manuscript provide a valid rationale for the proposed study, with clearly identified and justified research questions?

Reviewer #1: Yes

2. Is the protocol technically sound and planned in a manner that will lead to a meaningful outcome and allow testing the stated hypotheses?

Reviewer #1: Yes

3. Is the methodology feasible and described in sufficient detail to allow the work to be replicable?

Reviewer #1: Yes

4. Have the authors described where all data underlying the findings will be made available when the study is complete?

Reviewer #1: Yes

5. Is the manuscript presented in an intelligible fashion and written in standard English?

Reviewer #1: Yes

6. Review Comments to the Author

You may also provide optional suggestions and comments to authors that they might find helpful in planning their study.

Reviewer #1: Thank you for your response to my queries and comments. I have no further comments for the authors.

7. PLOS authors have the option to publish the peer review history of their article (what does this mean?). If published, this will include your full peer review and any attached files.

Reviewer #1: **Yes: **Dr Edith Holloway

---

## [Editor Report · Acceptance letter]

2 Jan 2024

PONE-D-23-19617R1 

PLOS ONE

Dear Dr. Dike, 

I'm pleased to inform you that your manuscript has been deemed suitable for publication in PLOS ONE. Congratulations! Your manuscript is now being handed over to our production team.

Kind regards, 

on behalf of

Dr. Muhammad Shahzad Aslam 

Academic Editor

PLOS ONE